# Heart rate variability analysis for the assessment of immersive emotional arousal using virtual reality: Comparing real and virtual scenarios

Javier Marín-Morales[1]*, Juan Luis Higuera-Trujillo[1], Jaime Guixeres[1], Carmen Llinares[1], Mariano Alcañiz[1], Gaetano Valenza[2]

1 Instituto de Investigación e Innovación en Bioingeniería, Universitat Politècnica de València, València, Spain, 2 Bioengineering and Robotics Research Centre E Piaggio & Department of Information Engineering, University of Pisa, Pisa, Italy

☯ These authors contributed equally to this work.
* jamarmo@i3b.upv.es

**Data Availability Statement:** The are ethical restrictions on making the data publicly available. The informed consent forms signed by the

## Abstract

Many affective computing studies have developed automatic emotion recognition models, mostly using emotional images, audio and videos. In recent years, virtual reality (VR) has been also used as a method to elicit emotions in laboratory environments. However, there is still a need to analyse the validity of VR in order to extrapolate the results it produces and to assess the similarities and differences in physiological responses provoked by real and virtual environments. We investigated the cardiovascular oscillations of 60 participants during a free exploration of a real museum and its virtualisation viewed through a head-mounted display. The differences between the heart rate variability features in the high and low arousal stimuli conditions were analysed through statistical hypothesis testing; and automatic arousal recognition models were developed across the real and the virtual conditions using a support vector machine algorithm with recursive feature selection. The subjects' self-assessments suggested that both museums elicited low and high arousal levels. In addition, the real museum showed differences in terms of cardiovascular responses, differences in vagal activity, while arousal recognition reached 72.92% accuracy. However, we did not find the same arousal-based autonomic nervous system change pattern during the virtual museum exploration. The results showed that, while the direct virtualisation of a real environment might be self-reported as evoking psychological arousal, it does not necessarily evoke the same cardiovascular changes as a real arousing elicitation. These contribute to the understanding of the use of VR in emotion recognition research; future research is needed to study arousal and emotion elicitation in immersive VR.

## Introduction

The study of emotions is a very important research topic for the understanding of human behaviour, as well as human perception, decision-making, creativity, memory and social

subjects prevent data from being publicly available for some years to come, even if the data is anonymized. Thus, should researchers wish to solicit the data privately, they can be requested via email, upon verification of all ethical aspects, at: i3b-instituto@upv.es.

**Funding:** The research leading to these results has received partial funding from the European Commission (Project HELIOS H2020-825585 and Project EXPERIENCE H2020-101017727), from the Universitat Politècnica de València (PAID-10-20), and from the Italian Ministry of Education and Research (MIUR) ("Department of Excellence" CrossLab project for the Univ. of Pisa). The funders had no role in study design, data collection and analysis, decision to publish, or preparation of the manuscript.

**Competing interests:** The authors have declared that no competing interests exist.

interaction. For many years, affective computing exploited knowledge derived from psycho-physiology, computer science, biomedical engineering, and artificial intelligence to develop systems that can recognise, model, and express emotions [1], with applications in healthcare [2], education [3] and entertainment [4]. An automatic emotional recognition system exploit implicit, maybe physiological measurements, in combination with machine-learning algorithms. There are three phases in the development of emotion recognition models: emotional modelling, emotion elicitation, and emotion recognition.

As for the emotional modelling, dimensional models may be used to describe the emotions as a multidimensional space where each dimension represents a fundamental property common to all emotions. The circumplex model of affect is one of the most commonly used dimensional models in affective computing; it uses three dimensions to model emotions: Arousal that represents the intensity of emotions in terms of activation from low to high; valence that is the degree to which an emotion is perceived as positive or negative; dominance also ranges from feelings of total lack of control, or influence on events and surroundings, to the opposite extreme of feeling influential and in control [5].

The elicitation of affective states is a challenging process and represents a critical stage in the process as conclusions obtained in lab condition are based on the assumption that the emotions evoked by the stimuli presented are similar to those evoked in the real-world [6]. The elicitation methods are grouped as active and passive. Active methods directly influence subjects, including behavioural manipulation [7], social interaction [8] and dyadic interaction [9]. On the other hand, passive methods use external audio-visual stimuli to elicit emotions. A widely used passive method is liked to the International Affective Picture System (IAPS), which is a large dataset of images of people, objects and events, rated in terms of arousal, valence and dominance [10]. In addition, many researches have used audio [11], music [12] and films to induce specific arousal and valence levels [13]. However, these passive emotion elicitation methods have two important limitations. First, the devices usually provide 2D stimuli, which may evoke low levels of presence. Presence is the feeling of "being there" when a virtual stimulus is presented, and it is an important indicator of the simulation's reliability when we evoke emotions using passive audio-visual stimuli [14]. Second, the majority of the stimuli are non-interactive, that is, the subjects are not able to intervene in the scene, which limits the simulation and analysis of interactive daily real-world tasks.

Virtual reality (VR) has recently started to be used as an emotion elicitation method in affective computing research, as it can contribute to overcome several limitations [15]. The popularity of VR has increased exponentially in recent years due to the development of a new generation of head-mounted displays [16]. These are fully immersive and interactive systems that isolate the user from external world stimuli and provide a complete simulated stereoscopic experience responsive to head movements, which in turn provokes a high sense of presence, that is, the strong illusion of being in the simulated environment [14]. Recent technological improvements of HMDs in terms of resolution and field of view are increasing their application in many research areas, including affective computing. In particular, arousal has been widely analysed in VR studies [15]. Jang et al. (2002) used a 3D virtual flight and driving simulator, suggesting the Heart Rate Variability low frequency / high frequency ratio as an objective measure of participants' arousal [17]. Meehan et al. (2005) analysed a 3D training experience and a pit room, correlating heart rate with presence levels in arousal environments [18]. Parsons et al. (2013) evoked arousal using a 3D high-mobility wheeled vehicle in a Stroop task, showing that high threat areas caused shorter interbeat intervals than low threat areas [19]. McCall et al. (2015) used a 3D room with threatening stimuli, such as explosions, spiders and gunshots, and correlated heart rate time-series with retrospective arousal ratings [20]. These experiments support the use of VR to evoke and analyse changes in the arousal dimension.

On the emotion classification, implicit measurements based on physiological signals may be used to analyse and automatically recognise the emotional responses of subjects and to classify emotions. Heart Rate Variability (HRV) series are widely used to gather implicit measurements to recognise arousal as they provide unique and non-invasive assessment tools of autonomic nervous system (ANS) control on cardiovascular dynamics, which change during different affective states [10]. HRV changes are regulated by the synergistic action of the two branches of the autonomous nervous system, that is, the sympathetic and parasympathetic nervous systems. HRV usually is analysed in the time, frequency, and non-linear domains. The majority of studies that have used HRV analysis in combination with arousal and immersive VR include time-domain features, in particular heart rate. For example, Breuninger et al. (2017) analysed arousal during a car accident [21], and Kisker et al. (2019) analysed the arousal response to a 3D exposure to a high height [22]. Some studies have also included frequency domain features, which may be related to the dynamics of the sympathetic and parasympathetic systems [21]. While the low frequency (LF) band reflects both sympathetic and vagal oscillations, HRV oscillations in the high frequency (HF) band may exclusively be linked to cardiac parasympathetic control [23–25]. Bian et al. (2016) analysed arousal in a 3D flight simulator [26], Zou et al. (2017) analysed arousal during a 3D fire evacuation, and Chittaro et al. (2017) analysed arousal in a comparison between a cemetery and a park. Finally, some studies have exploited HRV non-linear features in VR [27], as they have been shown to play a crucial role in affective state recognition [10]. Independently of the features, the majority of the studies that have used HRV and immersive VR relied on classic statistical methods such as hypothesis testing and correlations [15]; however, automatic arousal recognition models have recently been recommended using machine-learning algorithms, which allows to discriminate between states at a single-subject level. Marín-Morales et al. (2018) recognised arousal in architectural spaces [27], Granato et al. (2020) recognised arousal in a 3D video game [28], and Bălan et al. (2020) recognised fear in acrophobia therapy.

However, to extrapolate the insights obtained during arousing elicitation in a computer-simulated environment it is important to analyse the validity of the technology. This relates to the capacity to evoke a response from the user equal to the one that might be evoked by a real physical environment [29]. The assessment of the validity of the VR is critical in the analysis of physiological and behavioural dynamics during arousal elicitation. Few studies have performed direct comparisons between real and virtual stimuli; the majority have focused on psychological or behavioural responses. Chamilothori, Wienold, and Andersen, (2018) analysed the subjective perception of real and virtual daylit spaces, and found no significant differences between them in terms of self-assessment [30]. Heydarian et al. (2015) compared user performance in office-related activities such as reading text, and showed that they performed similarly in all measures in tasks in the virtual and the real-life environment [31]. Marín-Morales et al. (2019) compared navigation paths in free exploration in a real and a virtual museum, and showed differences during the first 2 minutes of the explorations [32]. On the other hand, the direct comparison of physiological responses is still an open issue. Higuera-Trujillo, Lopez-Tarruella and Llinares (2017) showed correlations in EDA responses in a comparison between a real and a virtual retail store. In addition, Marín-Morales et al. (2019) analysed the arousal responses between a real and a virtual museum and reported no differences in terms of self-assessment, but differences in brain dynamics [33].

To the best of our knowledge, cardiovascular dynamics in arousing VR environment has not been studied yet through a direct comparison between real and virtual environments.

In the present study we analyse arousal-induced responses during a free exploration of a real museum and their virtualisation displayed through an HMD. First, we analysed if the virtual environment evoked different levels of arousal in terms of subjects' perceptions. Next, we

performed a direct comparison of cardiovascular dynamics including in the time, frequency and non-linear domains. Statistical hypothesis tests of HRV features were performed between the high and low arousal stimuli, in both the real and the virtual conditions. In addition, a support vector machine classifier was developed to recognise arousal levels in both experimental conditions; this included a recursive feature elimination wrapper to explore the importance of each feature.

## Materials and methods

### Participants

A set of 60 healthy subjects (age 28.9 ± 5.44, 60% female) was recruited for the study; they were randomly assigned to the real or virtual museum scenarios. The following were the criteria to participate in the study: aged between 20 and 40 years; Spanish nationality; not suffering from cardiovascular nor obvious mental pathologies; not having formal education in art or a fine-art background; not having any previous virtual reality experience; and not having previously visited this particular art exhibition. To assess their mental health the participants were screened using a Patient Health Questionnaire (PHQ) [34]; a score of 5 or more caused the potential participants to be rejected. No subjects showed depressive symptoms. All methods and experimental protocols were approved by the ethics committee of the Universitat Politècnica de València. Written informed consent was obtained from all participants involved in the experiment, which allows us to publish the case details. The individual in this manuscript has given written informed consent (as outlined in PLOS consent form) to publish these case details.

### Physical museum exhibition

The art exhibition "Départ-Arrivée" by Christian Boltanski was selected to evoke an emotional experience in the wild. The exhibition was located in the Institut Valencià d'Art Modern (IVAM). The topic of the exhibition was the Nazi Holocaust; it consisted of exhibits displayed over 5 sequential rooms, with a total area of 750 m2. The participants were asked to freely explore the museum. The only specific instruction they received was that they were asked to observe in detail the 3 pieces of art in the last room. When the subjects arrived at the exit door, they were collected by the researcher, who at no time interfered with the exploration. The exhibition was divided into 8 independent stimuli, 5 rooms and 3 pieces of art (Fig 1). Following the explorations the subjects were asked to evaluate the stimuli using the Self- Assessment Manikin (SAM) questionnaire (from -4 to +4). The spatial positions of the subjects were

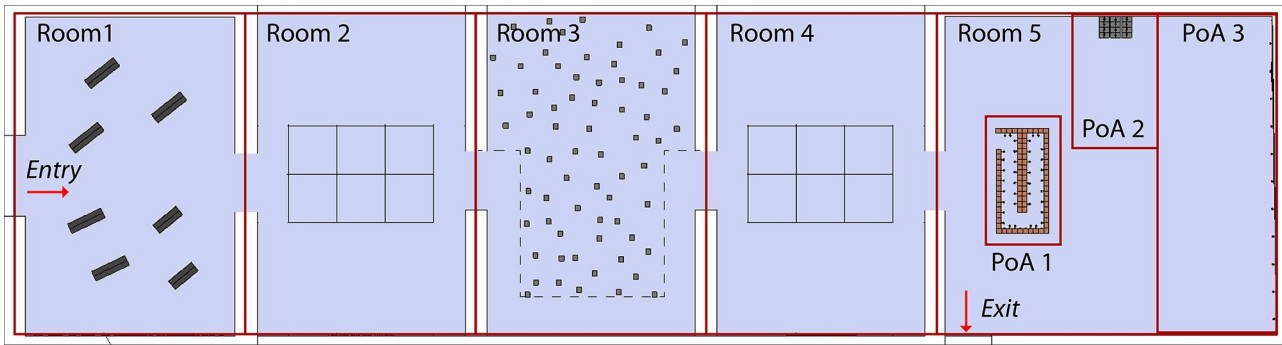

**Fig 1. Plan of the art-exhibition with the 5 rooms and 3 pieces of art.**

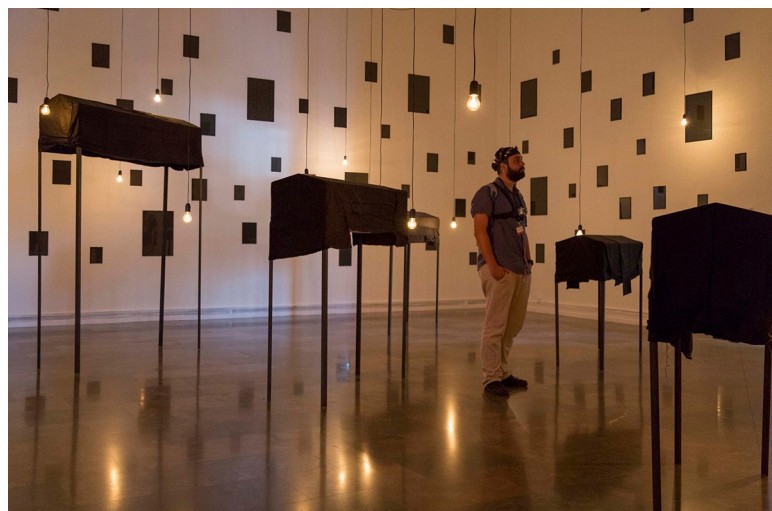

**Fig 2. Example of the experimental set-up in the real museum.**

tracked to synchronise the physiological signals with the stimuli. As part of this process, a GoPro camera was attached to the subjects' chests (using a harness) during the exploration. In addition, the subjects carried a backpack which contained a laptop that recorded the signals (Fig 2). The detailed methods to process and synchronise the navigation data were described in [32]. The subjects were asked, also, to evaluate the noise emitted by the sensors: "During the test, did you feel annoyed by the sensors?". No subjects reported "moderately", or "a lot".

## Virtual museum exhibition

A 3D VR simulation of the exhibition was developed using Unity 5.1 game engine software (www.unity3d.com) to try to recreate the same emotional experiences in the laboratory environment as had been evoked in the real museum. A spatial representation of the museum was developed using Rhinoceros v5.0. To provide the maximum level of realism the textures were partially extracted using photographs of the real environment. The development process involved architects who supervised the modelling of the environment, adjusting parameters, such as light, to match the real environment. Further information about the virtualisation was reported in [33]. An HTC Vive HMD was used to display the scenario. This has a resolution of 2160x1200 pixels (1080×1200 per eye) and a field of view of 110 degrees working at 90Hz refresh rate. The subjects, wearing the HMD and a joystick, were tracked in an area of 2x2 metres using two HTC base stations.

To navigate in the exhibition, a joystick-based teleport navigation metaphor included in the HTC Vive was used; this included a threshold of 2.5 metres as the maximum teleport radio to avoid large jumps. A PC Predator G6 (www.acer.com) via DisplayPort 1.2 and USB 3.0 was used to display the environment smoothly, that is, without interruptions. Fig 3 shows the experimental set up of the virtual museum, and Fig 4 shows a comparison between a photograph of the real museum and a screenshot of the virtual museum.

Before exploring the exhibition the subjects underwent training in a neutral environment to familiarise themselves with the technology, including the head-mounted display and the navigation metaphor. The training was not time limited; when it was completed the subjects were asked to explore the virtual museum following the same instructions as for the real museum. A visual feedback of the user's view was displayed on an external screen, and the

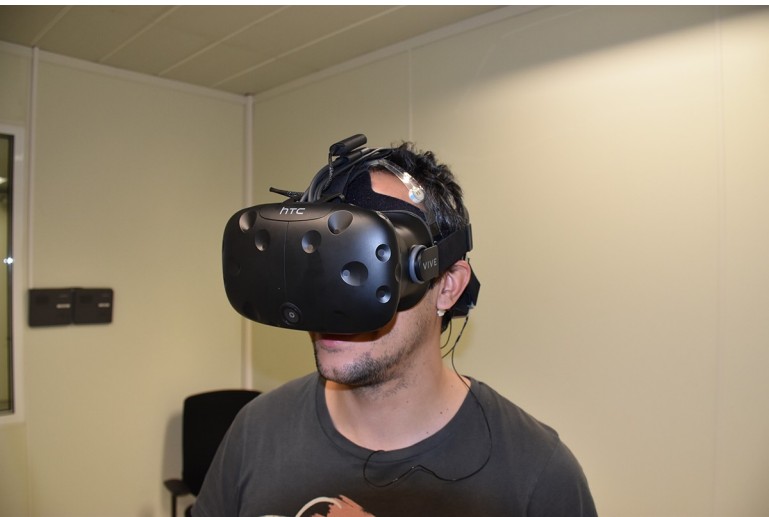

**Fig 3. Example experimental setup in the virtual museum.**

researchers stopped the recording and removed the HMD when the subjects arrived at the exit. Following the same methodology as for the real museum, the subjects were asked to evaluate the 8 stimuli using SAM questionnaires. The clear majority of the stimuli presented no self-assessment differences between the real and the virtual conditions [7].

## Biosignal processing

The electrocardiographic (ECG) signals were acquired during the free exploration using a B-Alert x10 device (Advanced Brain Monitoring, Inc. USA) sampled at 256. The left lead was located on the lowest left rib and the right lead on the right collarbone. Data from 15 subjects (7 from the real and 8 from the virtual museum) were rejected due to poor recording quality. The ECG signals were synchronised with the navigation, and then divided into independent signals for each stimulus based on the positions of the subjects. As the subjects could move backward and forward freely in the exhibition, they could visit the stimuli multiple times. In this case, their longest visit to any one stimulus was used for the HRV measurement. Signals less than 40 seconds duration were rejected. An analysis of the navigation was detailed in [32].

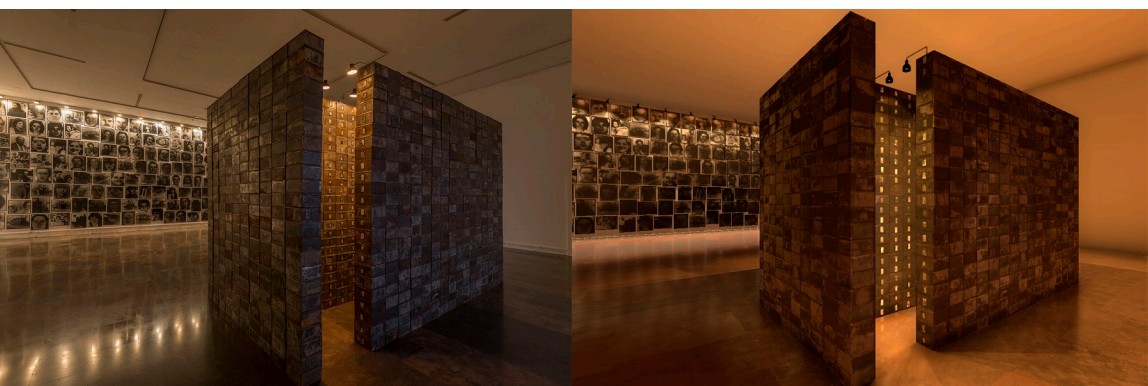

**Fig 4. Comparison between the real museum (left) and the virtual museum (right).**

An RR series was extracted from the ECG signal using the Pan-Tompkins's algorithm for QRS complex detection, and the smoothness prior detrending method was used to remove individual trends [35]. Kubios HRV software was used to correct artefacts and ectopic beats [36]. HRV standard analysis involving time and frequency domains was applied to the RR series. Moreover, we applied other HRV measures to quantify the nonlinear and complex dynamics [23]. The complete set of metrics is shown in Table 1.

Time-domain analysis includes average and standard deviation of RR intervals, the root mean square of successive differences of intervals (RMSSD) and the number of successive differences of intervals which differ by more than 50 ms (pNN50). In addition, we included the triangular interpolation of the HRV histogram and the baseline width of the RR histogram evaluated through triangular interpolation (TINN). To obtain the frequency domain features, the power spectral density (PSD) was calculated using Fast Fourier Transform and three bands: VLF (very low frequency, <0.04 Hz), LF (low frequency, 0.04–0.15 Hz) and HF (high frequency, 0.12–0.4 Hz). The peak value corresponding to the frequency having maximum magnitude and the power of the frequency band (in absolute and percentage terms) was calculated. Moreover, for the LF and HF bands, the normalised power (n.u.) was calculated as the percentage of the signals, subtracting the VLF from the total power, and the LF/HF ratio was calculated to quantify sympathovagal balance and to reflect sympathetic modulations. Finally, the total power was included. The VLF band was excluded from the analysis as it reflects changes due to slow regulatory mechanisms (e.g., thermoregulation) [23]. Moreover, Poincaré plot analysis was used; this is a quantitative-visual technique that summarises information about non-linear dynamics and detailed beat-to-beat data on heart behaviour and categorises them into functional classes: SD1, which is related with fast beat-to-beat variability, and SD2, which describes long-term variability [23].

Many nonlinear analyses have also been performed as it has been shown they are important quantifiers of cardiovascular control dynamics mediated by the ANS in affective computing [10]. Two measures of entropy were also included: sample entropy (SampEn), which provides an evaluation of time-series regularity [37], and Approximate Entropy, which detects changes in underlying episodic behaviour not reflected in peak occurrences or amplitudes [38]. DFA correlations are divided into short-term and long-term fluctuations: $\alpha1$ represents the fluctuation in the range of 4–16 samples, and $\alpha2$ refers to the range of 16–64 samples [39]. Finally, the D2 feature measures the complexity of the time series, providing information on the minimum number of dynamic variables needed to model the underlying system [40].

## Statistical analysis and machine learning

First, we bipolarised the subjects' self-assessments into high ($>0$) and low ($< = 0$) to determine the arousal of each stimulus. The self-assessments of the two arousal conditions were analysed

**Table 1. List of HRV features included in the analysis.**

| Time domain | Frequency domain | Other |
|---|---|---|
| Mean RR | LF peak | Poincaré SD1 |
| Std RR | HF peak | Poincaré SD2 |
| RMSSD | LF power | Approximate Entropy (ApEn) |
| pNN50 | LF power % | Sample Entropy (SampEn) |
| RR triangular index | LF power n.u. | DFA $\alpha1$ |
| TINN | HF power | DFA $\alpha2$ |
| | HF power % | Correlation dimension (D2) |
| | HF power n.u. | |
| | LF/HF power | |
| | Total power | |

using non-parametric hypothesis testing to determine if both environments were able to evoke arousal oscillations during the free exploration (based on subjects' perceptions). In addition, the HRV features were compared between the two arousal conditions in both environments using non-parametric hypothesis testing; this showed if there were differences in cardiovascular responses based on statistical inference. Moreover, two automatic arousal recognition models were created for the real and the virtual museums to explore the ability of HRV to discriminate between arousal states. The algorithm used was a support vector machine (SVM) pattern recognition [41]. The model was fed with the 23 HRV features calculated and the bipolarised arousal self-assessment, and calibrated using a leave-one-subject-out (LOSO) cross-validation procedure. Within the scheme the training set was normalised by subtracting the median value and dividing it by the mean absolute deviation over each dimension. In each iteration the validation set consisted of the cardiovascular responses of one specific subject, which were thereafter normalised using the median and deviation of the training set. The algorithm was optimised using a sigmoid kernel function in combination with a set of hyperparameters. In particular, we implemented a grid search, using a vector of the cost and gamma parameters with 15 values logarithmically spaced between 0.1 and 1000. The optimisation of the hyperparameters was performed with the objective of maximising Cohen's *kappa*, as the dataset was slightly unbalanced. In addition, we implemented a recursive feature elimination (RFE) to analyse the importance of each feature, selecting the variables that provided valuable information to extract the patterns. This was implemented in a wrapper approach, that is, it was performed on the training set of each fold, computing the median rank for each feature over all folds. Specifically, we used a recently developed nonlinear SVM-RFE which includes a correlation bias reduction strategy in the feature elimination procedure [42]. The performance of each model was evaluated using accuracy, Cohen's *kappa*, the ROC curve, AUC, the true positive rate (TPR) and the true negative rate (TNR). The algorithms were implemented using Matlab© R2018a, in combination with LIBSVM95.

## Results

### Subjects' self-assessment

As to the subjects' perceptions, Fig 5 shows the self-assessment scores in the high and low arousal conditions in both the real and virtual museums. The final number of participants was 23 in the real museum and 22 in the virtual museum. Due to the non-Gaussianity of the data (p < 0.05 from the Shapiro-Wilk test with a null hypothesis of having a Gaussian sample), Wilcoxon signed-rank tests were applied to the high vs low conditions in both museums. In the real museum, high arousing stimuli showed a mean arousal of 2.12 (σ = 1.01), whereas low arousing stimuli showed -1.31 (σ = 1.09). In the virtual museum, high arousing stimuli showed an average score of 2.51 (σ = 1.09), while low arousing stimuli showed -1.55 (σ = 1.31). These showed statistically significant arousal differences in both cases (p<0.001).

As to the physiological responses, Table 2 presents the results for the real museum, including the mean and standard deviations of each HRV feature for high and low arousal stimuli. Due to the non-Gaussianity of the data (p < 0.05 from the Shapiro-Wilk test with a null hypothesis of having a Gaussian sample), Wilcoxon signed-rank tests were applied. LF power % and LF power n.u. showed an increase in LF activity with low arousal stimuli, in combination with a higher LF peak. On the other hand, HF power, HF power % and HF power n.u. showed an increase in HF activity in the high arousal condition. In addition, the LF/HF ratio also showed an increase in vagal activity during visualisation of aversive high arousal stimuli. No differences were found in the real museum in time- or non-linear domain features. Table 3 shows the same analysis for the virtual museum condition. No features presented significant differences.

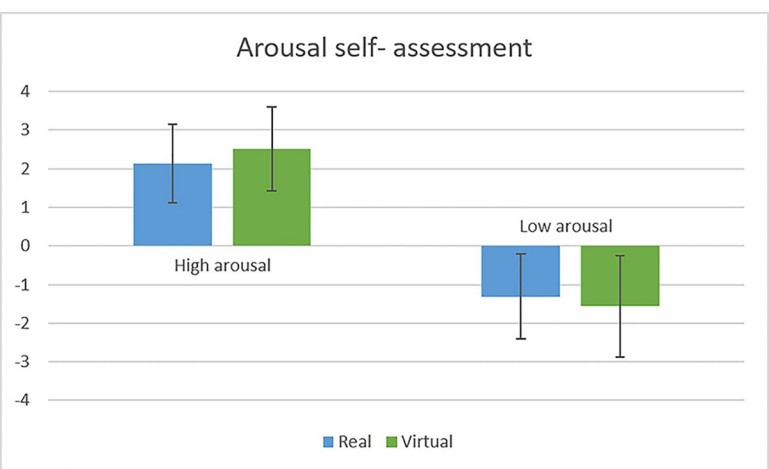

**Fig 5. Mean and standard deviations of the self-assessment scores in the real and virtual museums using SAM and a Likert scale (between -4 and +4), in both the high and the low arousal conditions.**

## Arousal recognition classification

Table 4 shows the performance of the arousal recognition models in both the real and the virtual museum conditions. In the real museum the model achieves 72.92% accuracy, being balanced in TPR (67.24%) and TNR (76.74%). The model has a score of 0.439 for Cohen's *kappa*.

**Table 2. HRV responses in the real museum in terms of arousal levels.**

| Feature | High arousal | Low arousal | p-value |
|---|---|---|---|
| Mean RR (sec) | 0.8184 (0.09890) | 0.7937 (0.08296) | 0.177 |
| Std RR (sec) | 0.0491 (0.02512) | 0.0560 (0.06003) | 0.575 |
| RMSSD (sec) | 0.0419 (0.02191) | 0.0361 (0.02198) | 0.055 |
| pNN50 (%) | 17.4386 (15.81660) | 13.5822 (14.06570) | 0.167 |
| HRV_tri_ind | 9.7107 (2.74080) | 9.4568 (3.66130) | 0.144 |
| TINN (sec) | 0.2166 (0.10265) | 0.2122 (0.11147) | 0.612 |
| LF peak (Hz) | 0.0733 (0.02738) | 0.0822 (0.02472) | 0.027 (*) |
| HF peak (Hz) | 0.2390 (0.07677) | 0.2217 (0.07148) | 0.14 |
| LF power (sec2) | 0.0017 (0.00296) | 0.0016 (0.00218) | 0.995 |
| LF power (%) | 55.7561 (19.43620) | 62.4636 (21.15690) | 0.019 (*) |
| LF power (n.u.) | 67.3262 (18.23250) | 76.7821 (14.31710) | 0.002 (**) |
| HF power (sec2) | 0.0006 (0.00067) | 0.0004 (0.00039) | 0.008 (**) |
| HF power (%) | 27.7152 (17.21820) | 19.6172 (14.56240) | 0.004 (**) |
| HF power (n.u.) | 32.5584 (18.18100) | 23.1467 (14.26570) | 0.002 (**) |
| LF/HF power | 4.0114 (5.10250) | 5.5624 (5.63240) | 0.002 (**) |
| Total power (sec2) | 0.0037 (0.01097) | 0.0049 (0.01497) | 0.455 |
| Poincaré SD1 (sec) | 0.0298 (0.01560) | 0.0257 (0.01568) | 0.061 |
| Poincaré SD2 (sec) | 0.0617 (0.03380) | 0.0738 (0.08448) | 0.768 |
| ApEn | 0.5968 (0.38241) | 0.5724 (0.38908) | 0.731 |
| SampEn | 1.2677 (0.80797) | 1.1588 (0.83422) | 0.281 |
| DFA alpha1 | 0.9354 (0.55931) | 0.9993 (0.62387) | 0.18 |
| DFA alpha2 | 0.3616 (0.31572) | 0.3696 (0.39586) | 0.771 |

Responses are reported using means and standard deviations. (*) and (**) indicate significant differences at p < 0.05 and p < 0.01, respectively (uncorrected).

**Table 3. HRV responses in the virtual museum in terms of arousal levels.**

| Feature | High arousal | Low arousal | p-value |
|---|---|---|---|
| Mean RR (sec) | 0.7654 (0.0882) | 0.7654 (0.0586) | 0.226 |
| Std RR (sec) | 0.0392 (0.0131) | 0.0355 (0.0109) | 0.094 |
| RMSSD (sec) | 0.0298 (0.0123) | 0.0258 (0.0082) | 0.091 |
| pNN50 (%) | 8.8690 (8.5668) | 5.9067 (6.8787) | 0.054 |
| HRV_tri_ind | 8.0030 (2.1301) | 7.8454 (2.2093) | 0.459 |
| TINN (sec) | 0.1688 (0.0661) | 0.1501 (0.0528) | 0.051 |
| LF peak (Hz) | 0.0888 (0.0206) | 0.0877 (0.0222) | 0.649 |
| HF peak (Hz) | 0.2348 (0.0744) | 0.2322 (0.0797) | 0.881 |
| LF power (sec2) | 0.0013 (0.0014) | 0.0012 (0.0012) | 0.682 |
| LF power (%) | 68.0645 (14.9879) | 71.6168 (16.8303) | 0.079 |
| LF power (n.u.) | 73.9998 (15.4908) | 76.5231 (17.2983) | 0.259 |
| HF power (sec2) | 0.0004 (0.0005) | 0.0003 (0.0003) | 0.181 |
| HF power (%) | 23.9533 (14.7126) | 22.0237 (16.7909) | 0.326 |
| HF power (n.u.) | 25.9334 (15.4641) | 23.4039 (17.2784) | 0.255 |
| LF/HF power | 4.7025 (4.5910) | 7.7323 (11.9998) | 0.255 |
| Total power (sec2) | 0.0018 (0.0020) | 0.0016 (0.0013) | 0.601 |
| Poincaré SD1 (sec) | 0.0212 (0.0087) | 0.0184 (0.0058) | 0.089 |
| Poincaré SD2 (sec) | 0.0509 (0.0171) | 0.0466 (0.0149) | 0.135 |
| ApEn | 0.4852 (0.3927) | 0.4154 (0.4215) | 0.327 |
| SampEn | 1.0680 (0.8270) | 0.9110 (0.8452) | 0.521 |
| DFA alpha1 | 0.8936 (0.6650) | 0.7977 (0.7242) | 0.847 |
| DFA alpha2 | 0.2366 (0.2171) | 0.1701 (0.1705) | 0.092 |

On the other hand, arousal recognition in the virtual museum had 70.39% accuracy, but with an unbalanced confusion matrix, in particular 46.51% for TPR. As to the confusion matrix and the data balance, the Cohen's *kappa* of the model was 0.265. Fig 6 shows the ROC curve of both models; arousal recognition in the real museum achieved an AUC score of 0.731, and 0.625 in the virtual museum.

Table 5 shows the feature ranking derived from the recursive feature elimination implemented with the support vector machine algorithm for both conditions. The model in the real condition used 6 features, the first 3 being from the frequency domain: HF peak, HF power n. u. and HF power %. In addition to these, 3 features from the non-linear domain were used in the model: SampEn, ApEn and DFA $\alpha 1$. The model in the virtual museum used 3 features: HF power %, HF peak and TINN.

## Discussion

In this study we investigated cardiovascular dynamics during high and low arousing elicitations through exploration of a real and a virtual museum and assessed the validity of VR by analysing physiological responses. An art exhibition about the Nazi Holocaust was simulated

**Table 4. Level of arousal recognition in both the real and the virtual museum conditions.**

| | | | | | | Confusion matrix | |
|---|---|---|---|---|---|---|---|
| Condition | Data balance (% high) | # Features selected | Acc. (%) | Kappa | AUC | TPR (%) | TNR (%) |
| Real Museum | 59.72% | 6 | 72.92 | 0.439 | 0.731 | 67.24 | 76.74 |
| Virtual Museum | 71.71% | 3 | 70.39 | 0.265 | 0.625 | 46.51 | 79.82 |

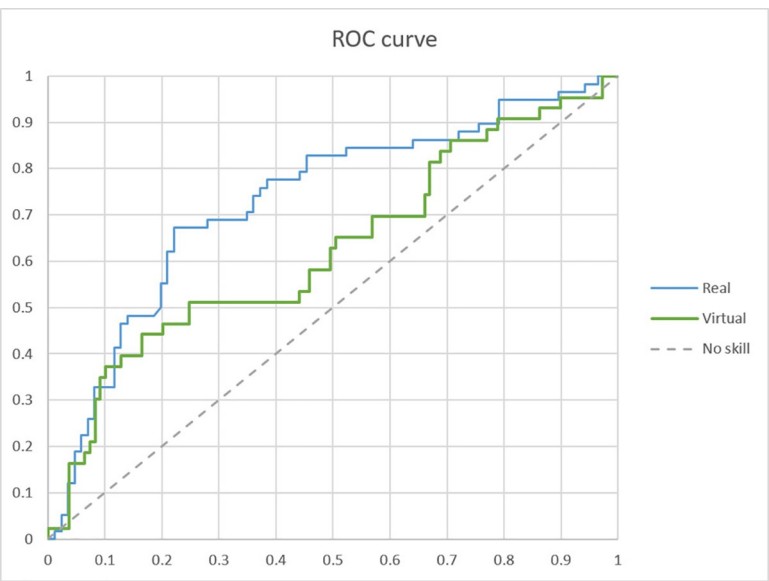

**Fig 6. ROC curve of the arousal recognition model in both the real and the virtual museum conditions.**

using a realistic 3D VR environment displayed through an HMD. To perform a direct comparison between the real museum and its virtualisation, 60 subjects were randomly assigned to perform a free exploration of one of the two conditions, real or virtual. The museums were divided into different areas that were self-assessed in terms of arousal, and HRV features were evaluated based on high-low arousal areas using a statistical hypothesis test and an SVM with RFE in both museums. While subjects' self-assessments in both the real and virtual museums showed different in perceived arousal levels, cardiovascular responses showed significant differences in the real museum, but not in its virtual simulation. The arousal recognition model in the real museum achieved good performance (*kappa* = 0.439) using frequency and non-linear domain features, but in the virtual condition the model did not achieve a good recognition (*kappa* = 0.265). These results suggest that, while the VR environment evoked similar psychological perceptions to those evoked in the real condition, it did not necessarily evoke the same autonomic responses as in real condition. These findings increase our understanding of VR in arousal recognition research and provide quantitative assessment tools for future studies.

More in detail, in the real museum, subjects' self-assessments showed differences between the high and low conditions during the free exploration. In terms of cardiovascular responses, the results obtained through the non-parametric statistical testing highlighted significant differences in sympathovagal responses in terms of LF peak, LF power (LF power % and LF

**Table 5. Feature ranking of the arousal recognition model in both the real and the virtual museum conditions.**

| | | Feature rankings | |
|---|---|---|---|
| | | **Real Museum** | **Virtual museum** |
| | 1 | HF peak | HF power % |
| | 2 | HF power n.u. | HF peak |
| | 3 | HF power % | TINN |
| | 4 | SampEn | |
| | 5 | ApEn | |
| | 6 | DFA $\alpha_1$ | |

power n.u.), HF power (HF power, HF power % and HF power n.u.) and LF/HF power. The increase in vagal activity during visualisation of high-arousal aversive emotional stimuli is in accordance with previous research. Sokhadze (2007) exposed students to a visual stimulation of mutilated bodies, and reported an increase in HF power and a decrease in LF/HF ratio [43]. Shenhav and Mendes (2014) exposed healthy participants to short film clips showing painful body injuries and found this evoked higher HF reactivity [44]. Garcia et al. (2016) showed an HF increment during the elicitation of negative emotions in depressive patients [45]. The exhibition included aversive stimuli (Nazi Holocaust content, including coffins and photographs of victims); our results showed that these high-arousal adverse stimuli increased vagal activity, and supported the use of HRV to detect arousal changes in the wild. However, the p-values of the statistical tests performed are not corrected for multiple comparisons and should be considered as a first exploratory step for the development of a multi-feature SVM for automatic arousal recognition.

In the use of a virtual museum, while subjects' self-assessment scores showed differences between different arosing explorations, no significant differences were found between the arousing conditions in terms of cardiovascular dynamics. Although the results showed the same trends for the real stimuli, that is, higher HF and lower LF/HF for high-arousal stimuli, statistical differences between arousing sessions were not identified. This could be due to the VR itself, which can produce an increase in the arousal perception, especially because subjects had had no previous experience with VR; however, inter-subject variability associated cardiovascular responses to VR stimuli may significantly bias the results due to the novelty of VR, so further studies are needed. Moreover, although the level of realism achieved in the museum was quite high, VR is necessarily unfamiliar to subjects to some extent [46]. In recent years, an uncanny-valley of the mind reaction has been found in avatar-based VR research, where the agents performed in similar, but not identical, ways to humans [47]. This uncanny-valley effect might be coming into play in this aversive virtual exhibition and may evoke a sense of eeriness that impacts on cardiovascular dynamics. In addition, the differences between the responses in VR and in the physical environment might be measured by sense of presence [14] and emotional embodiment [48], which could be integrated into the circumplex model of affects to describe the complete emotional experience in VR.

The performance of the arousal recognition models was in accordance with the results of the statistical tests. The arousal recognition in the real museum was good, achieving 72.92% of accuracy, a *kappa* score of 0.439 and an AUC score of 0.731, including a balanced confusion matrix. The model used a total of 6 features, the first three being from the frequency domain, HF peak, HF power n.u. and HF power %. Therefore, vagal activity seems the most important measurement for recognising arousal in the free exploration of a real museum. In addition, the non-linear domain (SampEn, ApEn and DFA $\alpha_1$) contributed to increase the recognition power, which is in accordance with previous research that showed the important role that non-linear dynamics play in affective identification [10]. On the other hand, the model did not achieve a balanced confusion matrix in the VR setting, returning a true positive rate of 46.31%. This impacts on the general performance, with a *kappa* of 0.265 and an AUC of 0.625, which can be considered poor as they are below 0.4 and 0.7, respectively. This effect can be seen graphically in Fig 6, where the ROC curve of the virtual museum is considerably below that of the real museum, and close to the no skill line.

Our results suggest that the classification of emotional arousal in VR is challenging using cardiovascular dynamics exclusively. These results are in accordance with previous research. Marín-Morales et al. (2018) recognised arousal in 360˚ immersive rooms, showing that the most important features were EEG-related, not HRV-related [27]. Moreover, Marín-Morales et al. (2019) presented an explorative analysis of the experiment undertaken in the present

study and showed that in the virtual museum EEG is the more important signal [33]. In addition, Granato et al. (2020) and Bălan et al. (2020) presented two multi-signal emotion recognition models that showed that EDA features are more effective than HRV in discriminating arousal and fear, respectively, in VR [28, 49]. However, a limitation of these studies is that they used only time-domain features. Although previous research has shown statistically significant differences in arousal levels using hypothesis testing on HRV features [50], to the best of the authors' knowledge no arousal recognition model has effectively assessed arousal dynamics in immersive VR using only HRV features [50]. Therefore, it has been shown that EDA is more effective than HRV for analysing ANS arousal related dynamics in VR [28], and that the CNS dynamics captured by EEG can recognise arousal states, brain synchronisation features being particularly effective [33]. Considering that HRV was previously very effective in recognising arousal using 2D stimuli such as IAPS [10], more research is needed to continue analysing and thoroughly understanding the cardiovascular oscillations in VR, as to date very few studies have developed arousal recognition models that go beyond classic statistical methods, and the present study is the first direct comparison that includes physical space as a benchmark. Moreover, knowledge of HRV-related changes at different levels of arousal, both in a virtual and real environment, is relevant because of the widespread use of smartwatches and other wearable devices that are able to ubiquitously monitor cardiovascular variability in an ecological fashion. In addition, baseline data gathered during non-emotional elicitation should be investigated for further normalization steps aimed at improving the accuracy of the proposed computational models.

Immersive VR-based human emotion research has grown exponentially in recent years due to VR's ability to simulate environments in laboratory conditions. Therefore, reaching a profound understanding of physiological dynamics during arousal oscillations is a critical point in the validation of VR technologies as emotion elicitation. The replication of affective computing experiments previously developed in 2D, but using immersive stimuli, will increase the understanding about the relationship between presence and emotion. It is, thus, important to go further than classic statistical testing, by combining implicit measures with machine-learning algorithms to model the patterns behind physiological responses, which often present non-linear relationships [10]. In addition, direct comparisons between real environments and their VR simulations will provide benchmarks from which to compare the insights obtained from VR, and will be one of the keys in the validation of immersive simulated stimuli, and in understanding their differences from, and similarities to, physical reality. As the use of VR in emotion recognition research continues to mature, it will provide new opportunities for affective computing research. First, it can open the possibility of studying new dimensions such as dominance, as the sense of control of one's environment is very difficult to simulate using non-immersive stimuli. For example, a human can feel disgust when (s)he sees a snake in 2D, but it is difficult to feel insecurity or fear without a high sense of presence in the simulated environment, where immersion plays a critical role [14]. Therefore, VR might help analyse emotional states such as fear and sense of security. In addition, VR provides a level of interactivity that will help affective computing research to simulate and analyse daily tasks, thus helping to narrow the gap between laboratory elicitation and real-world situations. In particular, one of the most important interactions for future research is social interaction, since it is very related to emotion regulation abilities [51]. Recent technological developments are opening the possibility of creating realistic avatars that, in combination with improvements in chat-bots, will allow researchers to recreate controlled naturalistic conversations in VR settings. The synergies between biomedical engineering, VR and artificial intelligence might in future years revolutionise the application of affective computing to many research areas, such as health, psychology, management, architecture, marketing and neuroeconomics.

## Author Contributions

**Conceptualization:** Javier Marín-Morales, Juan Luis Higuera-Trujillo, Carmen Llinares.

**Data curation:** Javier Marín-Morales, Gaetano Valenza.

**Formal analysis:** Javier Marín-Morales, Gaetano Valenza.

**Funding acquisition:** Mariano Alcañiz.

**Methodology:** Javier Marín-Morales, Juan Luis Higuera-Trujillo, Carmen Llinares.

**Project administration:** Javier Marín-Morales.

**Resources:** Juan Luis Higuera-Trujillo.

**Supervision:** Jaime Guixeres, Carmen Llinares, Mariano Alcañiz, Gaetano Valenza.

**Writing – original draft:** Javier Marín-Morales.

**Writing – review & editing:** Javier Marín-Morales, Juan Luis Higuera-Trujillo, Jaime Guixeres, Carmen Llinares, Mariano Alcañiz, Gaetano Valenza.

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
