## [Decision Letter · Decision Letter 0]

11 Feb 2021

PONE-D-20-32832

Heart rate variability analysis for the assessment of immersive emotion elicitation using virtual reality: Comparing real and virtual scenarios in the arousal dimension

PLOS ONE

Dear Dr. Marín-Morales,

Thank you for submitting your manuscript to PLOS ONE. After careful consideration, we feel that it has merit but does not fully meet PLOS ONE’s publication criteria as it currently stands. Therefore, we invite you to submit a revised version of the manuscript that addresses the points raised during the review process.

We look forward to receiving your revised manuscript.

Kind regards,

Leontios J. Hadjileontiadis

Academic Editor

PLOS ONE

Additional Editor Comments (if provided):

This manuscript seems highly overlapping with another (“Real vs. immersive-virtual emotional experience: Analysis of psycho-physiological patterns in a free exploration of an art museum”). It would have to be made very clear what the unique contributions of the current work are and the content of the previous work would need to be explained in more detail in the introduction.

Journal Requirements:

2) We note that you have indicated that data from this study are available upon request. PLOS only allows data to be available upon request if there are legal or ethical restrictions on sharing data publicly. For more information on unacceptable data access restrictions, please see http://journals.plos.org/plosone/s/data-availability#loc-unacceptable-data-access-restrictions.

3) We note that Figures 2 & 3 include an image of a participant in the study. 

Reviewers' comments:

Reviewer's Responses to Questions

**Comments to the Author**

1. Is the manuscript technically sound, and do the data support the conclusions?

Reviewer #1: Yes

Reviewer #2: Partly

2. Has the statistical analysis been performed appropriately and rigorously? 

Reviewer #1: No

Reviewer #2: I Don't Know

3. Have the authors made all data underlying the findings in their manuscript fully available?

Reviewer #1: No

Reviewer #2: No

4. Is the manuscript presented in an intelligible fashion and written in standard English?

Reviewer #1: Yes

Reviewer #2: Yes

5. Review Comments to the Author

Reviewer #1: The study looked at the differences between the heart rate variability features in the high and low arousal stimuli across the real and the virtual conditions using a support vector machine algorithm. The results are interesting for HRV research community but there are several major issues that need to be addressed.

1. Subjects’ demographics (age, BMI, gender, educational background etc) are missing.

2. Table for summary of self assessment questionnaire for real and visual environments is also missing.

3. In Tables 2&3 units of HRV features are missing.

4. I did not understand how SVM models were trained. Training/testing protocols with SVM parameters are missing. How do I know that best SVM model was used?

5. I would suggest authors to summarize the HRV results in figures.

6. inclusion and exclusion criteria should be clarified. In the HRV tables, did you include all 60 subjects' data? Please clarify

Reviewer #2: The current manuscript describes a study of heart rate variability as a potential maker of individuals’ ‘emotion’ when viewing a museum exhibit in physical reality versus virtual reality (VR). I feel that conducting this type of comparison between VR and real spaces is an important endeavor, and so appreciate this aim of the work. The authors apparently went to great lengths to make this as close of a comparison as possible which is a real strength of the project.

My biggest concern about this manuscript is that it appears very similar and highly overlapping with the previous paper: “Real vs. immersive-virtual emotional experience: Analysis of psycho-physiological patterns in a free exploration of an art museum”. Although the previous paper included both EEG and ECG whereas the current paper focuses on ECG with some slightly different approaches. The unique contributions of the current report would need to be very clearly laid out and the authors would need to very clearly state points of overlap versus departure from the previous manuscript to show the uniqueness and contribution of the current work. Why focus on only ECG when previous working using EEG performed better?

Other comments:

Conceptually, the authors discuss the evaluation of “emotion”, which as they note in the introduction can be thought of as a circumplex on three dimensions, of which arousal is one. Given that the current study only evaluates arousal, both in terms of self-report and in terms of what heart rate variability (HRV) can tell us, it does not seem appropriate to frame this as an evaluation of “emotion”. Rather, I suggest that the authors describe the study as an evaluation of arousal. This also suggests that the current study cannot be motivated by an introduction describing specific emotions when arousal is nonspecific by its nature.

In this context what does validation mean? Under what contexts would the procedure be determined to be ‘valid’? How does the machine learning piece contribute to validation? The AUCs weren’t very good, so what does this tell us about validation?

The authors list the many features that can be extracted from HRV and include all in their models, but they never describe what these features are understood to mean. Some description of the meaning and use of the various types of features would be very useful. How are these features considered different from one another such that they can all be valuable to assess? How overlapping are the various metrics?

I have questions about whether any baseline physiological data were collected and how those were integrated into the analyses. There are several differences in physical movement in addition to the novelty of the VR that could account for some of the differences between physiological data patterns between the two.

Was any statistical adjustment done for multiple comparisons? There are an awful lot of analyses run in this paper.

The authors seem to justify the investigation of VR as an emotional induction tool by suggesting that presence is required for evoking emotion, when clearly this is not the case (pg 4), in addition although the ability to simulate real-world tasks may be helpful for investigating some elements of emotion-related experience, clearly this is also not required (e.g., integral vs incidental emotion). Thus, I’m a little confused by the content on page 4. There are certainly other reasons why researchers may want to study emotion in the context of VR, so I don’t disagree that this is a worthy thing to investigate, I just find the reasoning provided to be questionable.

There are several sentences that don’t make sense to me. This includes:

Pg 3, line 46: “the applications of affective computing are impacting transversally…”

Pg 4, line 82: “The recent technological improvements in the performance of HMDs…” – how is HMD performance boosting emotion recognition?

6. PLOS authors have the option to publish the peer review history of their article (what does this mean?). If published, this will include your full peer review and any attached files.

Reviewer #1: **Yes: **Ahsan Khandoker

Reviewer #2: No

---

## [Author Response · Author response to Decision Letter 0]

22 Mar 2021

Statement of Changes and Answers to Comments

We would like to thank the editor and reviewers for their careful consideration of our paper and for their valuable suggestions as to how to improve it. We believe that this revised version of manuscript ref. number PONE-D-20-32832 addresses the criticisms/comments raised by the reviewers. The revised text is highlighted in red to easy the revision process.

STATEMENT OF CHANGES

Prompted by the reviewers’ suggestions, the main changes to the revised manuscript are as follows:

1) The study description now focuses on the arousal recognition rather than emotion recognition, and changes reflect in the Abstract, Introduction, Materials and methods, Results and Discussion sections.

2) The relationship between presence and emotion elicitation is now described in the Introduction section. 

3) Results on the self-assessment scores are now detailed in the Results section.

4) The final number of participant is now included in the Results section.

5) Table 2 and 3 now include the unit measure.

6) The importance of HRV analysis in arousal recognition is now empathised in the Discussion section.

7) Future research directions are now mentioned in the Discussion section.

8) Minor typos are now fixed.

 

Answer to reviewers

Reviewer #1

Comments:

Q1.1) The study looked at the differences between the heart rate variability features in the high and low arousal stimuli across the real and the virtual conditions using a support vector machine algorithm. The results are interesting for HRV research community but there are several major issues that need to be addressed.”. 

R1.1) Thank you for your comment. We worked hard to address all your concerns in the revised manuscript. Please see details below.

Q1.2) Subjects’ demographics (age, BMI, gender, educational background etc) are missing.

R1.2) The revised manuscript includes age and gender information of the subjects participating in the study, and subjects with educational background in fine art were excluded because of possible emotional bias for the presented stimuli. The information is included in the revised manuscript at line 150 as follows:

 “A set of 60 healthy subjects (age 28.9 ± 5.44, 60% female) was recruited for the study; they were randomly assigned to the real or virtual museum scenarios. The following were the criteria to participate in the study: aged between 20 and 40 years; Spanish nationality; not suffering from cardiovascular nor obvious mental pathologies; not having formal education in art or a fine-art background; not having any previous virtual reality experience; and not having previously visited this particular art exhibition. To assess their mental health the participants were screened using a Patient Health Questionnaire (PHQ) [35]; a score of 5 or more caused the potential participants to be rejected. No subjects showed depressive symptoms. “

Q1.2) Table for summary of self assessment questionnaire for real and visual environments is also missing.

R1.2) We believe that a Table would not be optimal to list the few values of the self-assessment tests. Following the reviewer’s suggestion, the self-assessment scores are now included in the text at line 299 as follows:

 “As to the subjects’ perceptions, Fig 5 shows the self-assessment scores in the high and low arousal conditions in both the real and virtual museums. The final number of participants was 23 in the real museum and 22 in the virtual museum. Due to the non-Gaussianity of the data (p < 0.05 from the Shapiro-Wilk test with a null hypothesis of having a Gaussian sample), Wilcoxon signed-rank tests were applied to the high vs low conditions in both museums. In the real museum, high arousing stimuli showed a mean arousal of 2.12 (σ=1.01), whereas low arousing stimuli showed -1.31 (σ=1.09). In the virtual museum, high arousing stimuli showed an average score of 2.51 (σ=1.09), while low arousing stimuli showed -1.55 (σ=1.31). These showed statistically significant arousal differences in both cases (p<0.001).”

Q1.3) In Tables 2&3 units of HRV features are missing.

R1.3) Following the reviewer’s suggestion, the revised manuscript now includes the measurement units for all features listed in Tables 2 and 3.

Q1.4) I did not understand how SVM models were trained. Training/testing protocols with SVM parameters are missing. How do I know that best SVM model was used?

R1.4) Thank you for your comment. The machine learning strategy and related training/testing phases are detailed in the revised manuscript at line 275 as follows:

 “The algorithm used was a support vector machine (SVM) pattern recognition [41]. The model was fed with the 23 HRV features calculated and the bipolarised arousal self-assessment, and calibrated using a leave-one-subject-out (LOSO) cross-validation procedure. Within the scheme the training set was normalised by subtracting the median value and dividing it by the mean absolute deviation over each dimension. In each iteration the validation set consisted of the cardiovascular responses of one specific subject, which were thereafter normalised using the median and deviation of the training set. The algorithm was optimised using a sigmoid kernel function in combination with a set of hyperparameters. In particular, we implemented a grid search, using a vector of the cost and gamma parameters with 15 values logarithmically spaced between 0.1 and 1000. The optimisation of the hyperparameters was performed with the objective of maximising Cohen’s kappa, as the dataset was slightly unbalanced. In addition, we implemented a recursive feature elimination (RFE) to analyse the importance of each feature, selecting the variables that provided valuable information to extract the patterns. This was implemented in a wrapper approach, that is, it was performed on the training set of each fold, computing the median rank for each feature over all folds. Specifically, we used a recently developed nonlinear SVM-RFE which includes a correlation bias reduction strategy in the feature elimination procedure [42].”

 In summary, we applied a LOSO cross-validation procedure using a SVM with a sigmoid kernel. The hyperparameter C and gamma were tuned using a grid search with 15 values logarithmically spaced between 0.1 and 1000. In addition, a recursive feature elimination was performed, and the selected features are included in Table 5.

Q1.5) I would suggest authors to summarize the HRV results in figures.

R1.5) Thank you for this suggestion. For the sake of conciseness, we would like to keep the results on HRV statistics as a Table format. Graphical representations, in fact, such as boxplots, would require a quite significant amount of space to show all the 23 HRV parameters. 

Q1.6) inclusion and exclusion criteria should be clarified. In the HRV tables, did you include all 60 subjects' data? Please clarify

R1.6) HRV data were eventually discarded from further analyses in case of poor quality as specified in the revised manuscript at line 222 as follows: 

 “Data from 15 subjects (7 from the real and 8 from the virtual museum) were rejected due to poor recording quality.” 

 Following the reviewer’s suggestion, the final number of participants is now included in the revised manuscript in the Results section at line 300 as follows: 

 “The final number of participants was 23 in the real museum and 22 in the virtual museum.”

 

Reviewer #2

Comments:

Q2.1) The current manuscript describes a study of heart rate variability as a potential maker of individuals’ ‘emotion’ when viewing a museum exhibit in physical reality versus virtual reality (VR). I feel that conducting this type of comparison between VR and real spaces is an important endeavor, and so appreciate this aim of the work. The authors apparently went to great lengths to make this as close of a comparison as possible which is a real strength of the project. 

R2.1) Thank you for your comments aimed at improving the quality of the study description and results. We worked hard to address all your concerns. Please see a detailed answer to your comments below.

Q2.2) My biggest concern about this manuscript is that it appears very similar and highly overlapping with the previous paper: “Real vs. immersive-virtual emotional experience: Analysis of psycho-physiological patterns in a free exploration of an art museum”. Although the previous paper included both EEG and ECG whereas the current paper focuses on ECG with some slightly different approaches. The unique contributions of the current report would need to be very clearly laid out and the authors would need to very clearly state points of overlap versus departure from the previous manuscript to show the uniqueness and contribution of the current work. Why focus on only ECG when previous working using EEG performed better?

R2.2) In the recently published manuscript entitled “Real vs. immersive-virtual emotional experience: Analysis of psycho-physiological patterns in a free exploration of an art museum” we performed an explorative analysis on brain and heartbeat dynamics with respect to emotions elicited through a real museum exploration or its virtualization. 

 Considering the current widespread use of smartwatches and other wearable devices able to continuously monitor HRV through pulse oximeters or portable ECGs, we do believe that a study focused on HRV dynamics exclusively is of great interest for the scientific community. Particularly, we focused the investigation on the development of an arousal recognition system based on HRV information exclusively. Following the reviewer’s suggestion, we now emphasize this point in the revised manuscript at line 448 as follows: 

 “Moreover, knowledge of HRV-related changes at different levels of arousal, both in a virtual and real environment, is relevant because of the widespread use of smartwatches and other wearable devices that are able to ubiquitously monitor cardiovascular variability in an ecological fashion.”

 Following the reviewer’s comment, here we also list major differences between the present and previous Plos One publication:

- The previous paper does not apply descriptive nor inferential statistics to features of cardiovascular dynamics between high and low arousing stimuli. Note that our results are in accordance with previous research on aversive stimuli, showing an increase in vagal activity during the visualisation of aversive arousing stimuli (Sokhadze, 2007; Garcia et al., 2016),

- The previous PloS One paper does not show the performance on the arousal recognition using only HRV features, which is indeed one of main goals of the present manuscript. For the first time, in the present manuscript we show results on the comparison between a physical and a virtual space for HRV features.

- The previous Plos One paper used a supervised machine learning pipeline in combination with a PCA-based feature reduction, thus the information carried by each HRV feature for an arousal recognition cannot be inferred. In the present paper, results show the information that the HRV features defined in the nonlinear domain have in the recognition of arousal, which is also in accordance with previous research in lab environment (Valenza et al. 2011).

- The take home message of the present paper is that, while both virtual and real emotional scenarios elicit different levels of arousing emotional stimuli, as demonstrated by the self-assessment scores, cardiovascular patterns associated with the physical museum are different from the ones elicited by the virtual museum. We believe this is an important result to be disseminated to the affective computing research community.

 In conclusion, many important novelties are with the current manuscript in terms of methods including a detailed characterization of HRV dynamics, but also in terms of new insights in the field of affective computing.

 Valenza, G., Lanata, A., & Scilingo, E. P. (2011). The role of nonlinear dynamics in affective valence and arousal recognition. IEEE transactions on affective computing, 3(2), 237-249.

 Sokhadze EM. Effects of music on the recovery of autonomic and electrocortical activity after stress induced by aversive visual stimuli. Appl Psychophysiol Biofeedback. Springer; 2007;32: 31–50.

Garcia RG, Valenza G, Tomaz C, Barbieri R. Relationship between cardiac vagal activity and mood congruent memory bias in major depression. J Affect Disord. Elsevier; 2016;190: 19–25.

Q2.3) Other comments:

 Conceptually, the authors discuss the evaluation of “emotion”, which as they note in the introduction can be thought of as a circumplex on three dimensions, of which arousal is one. Given that the current study only evaluates arousal, both in terms of self-report and in terms of what heart rate variability (HRV) can tell us, it does not seem appropriate to frame this as an evaluation of “emotion”. Rather, I suggest that the authors describe the study as an evaluation of arousal. This also suggests that the current study cannot be motivated by an introduction describing specific emotions when arousal is nonspecific by its nature.

R2.3) We agree with the reviewer that it is not appropriate to frame this study as an evaluation of emotion. Following the reviewer’s suggestion, the manuscript now focuses on arousal recognition.

 First, the Introduction section has been revised also highlighting the state of art related to arousal recognition and heart rate variability. Revised text now at line 82 is as follows:

 “In particular, arousal has been widely analysed in VR studies [15]. Jang et al. (2002) used a 3D virtual flight and driving simulator, suggesting the HRV low frequency / high frequency ratio as an objective measure of participants’ arousal [17]. Meehan et al. (2005) analysed a 3D training experience and a pit room, correlating heart rate with presence levels in arousal environments [18]. Parsons et al. (2013) evoked arousal using a 3D high-mobility wheeled vehicle in a Stroop task, showing that high threat areas caused shorter interbeat intervals than low threat areas [19]. McCall et al. (2015) used a 3D room with threatening stimuli, such as explosions, spiders and gunshots, and correlated heart rate time-series with retrospective arousal ratings [20]. These experiments support the use of VR to evoke and analyse changes in the arousal dimension.”

 In addition, many improvements have been done among the manuscript to frame the research in arousal recognition.

 Abtract:

 Line 27: “and automatic arousal recognition models were developed across the real and the virtual conditions”

 Introduction:

 Line 82: “arousal has been widely analysed in VR studies”

 Linea 94: “Heart Rate Variability (HRV) series are widely used to gather implicit measures to recognise arousal”

 Line 113: “however, some recent research has started to develop automatic arousal recognition models using machine-learning algorithms”

 Line 118: “However, to extrapolate the insights obtained during arousing elicitation in a computer-simulated environment it is important to analyse the validity of the technology.”

 Line 121. “The validity of the VR is critical in the analysis of physiological and behavioural dynamics during arousal elicitation”

 Line 134: “analysed the arousal responses between”

 Line 136: “In particular, cardiovascular dynamics in arousing VR has not been studied through a direct comparison between real and virtual environments”

 Materials and methods.

 Line 274: “two automatic arousal recognition models were created for the real and the virtual museums to explore the ability of HRV to discriminate between arousal states

 Results:

 Line 332: “Arousal Recognition Classification”

 Line 345: “ROC curve of the arousal recognition model in both the real and the virtual museum conditions”

 Line 355: “Feature ranking of the arousal recognition model in both the real and the virtual museum conditions”

 Discussion:

 Line 378: “These findings help to increase understanding of the use of VR in arousal recognition research”

 Line 414: “The performances of the arousal recognition models were in accordance with the results of the statistical tests”

 Line 439: “to the best of the authors’ knowledge no arousal recognition model has effectively assessed arousal dynamics in immersive VR using only HRV features”

 Line 456: “Therefore, reaching a profound understanding of physiological dynamics during arousal oscillations is a critical point”

Q2.4) In this context what does validation mean? Under what contexts would the procedure be determined to be ‘valid’? How does the machine learning piece contribute to validation? The AUCs weren’t very good, so what does this tell us about validation? 

R2.4) In the present manuscript, we proposed a multiparametric approach that combines cardiovascular variability features in order to discriminate emotional arousal levels. To this extent, we developed a Support Vector Machine model that properly combines the features trying to discriminate the arousal at a single-subject level using a non-linear transformation kernel. The AUC represents the performance of the model in terms of arousal recognition, and the model validation is related to the procedure detailed in the revised manuscript at line 275 as follows:

 “The algorithm used was a support vector machine (SVM) pattern recognition [41]. The model was fed with the 23 HRV features calculated and the bipolarised arousal self-assessment, and calibrated using a leave-one-subject-out (LOSO) cross-validation procedure. Within the scheme the training set was normalised by subtracting the median value and dividing it by the mean absolute deviation over each dimension. In each iteration the validation set consisted of the cardiovascular responses of one specific subject, which were thereafter normalised using the median and deviation of the training set. The algorithm was optimised using a sigmoid kernel function in combination with a set of hyperparameters. In particular, we implemented a grid search, using a vector of the cost and gamma parameters with 15 values logarithmically spaced between 0.1 and 1000. The optimisation of the hyperparameters was performed with the objective of maximising Cohen’s kappa, as the dataset was slightly unbalanced. In addition, we implemented a recursive feature elimination (RFE) to analyse the importance of each feature, selecting the variables that provided valuable information to extract the patterns. This was implemented in a wrapper approach, that is, it was performed on the training set of each fold, computing the median rank for each feature over all folds. Specifically, we used a recently developed nonlinear SVM-RFE which includes a correlation bias reduction strategy in the feature elimination procedure [42].”

. 

Q2.5) The authors list the many features that can be extracted from HRV and include all in their models, but they never describe what these features are understood to mean. Some description of the meaning and use of the various types of features would be very useful. How are these features considered different from one another such that they can all be valuable to assess? How overlapping are the various metrics?

R2.5) Thank you for your comment. HRV feature description are in different sections of the revised manuscript as reported in the paragraphs below; however, please note that a few HRV features only have a well-defined physiological correlates. In fact, while HF power is liked to the cardiac parasympathetic activity, and LF power is related to the sympathovagal activity, all other features defined in the time (mean, std, RMSSD, pNN50, HRV_tri_ind, TINN) and geometic and nonlinear domains (Poincaré SD1, Poincaré SD2, ApEn, SampEn, DFA alpha1, DFA alpha2) are linked to non-specific cardiovascular dynamics with no well-defined physiological correlate.

 Introduction:

 “The majority of studies that have used HRV analysis in combination with arousal and immersive VR include time-domain features, in particular heart rate. Two examples are: Breuninger et al. (2017) analysed arousal during a car accident [22], and Kisker et al. (2019) analysed the arousal response to a 3D exposure to a high height [23]. Some studies have also included frequency domain features, which describe the dynamics of the sympathetic and parasympathetic systems [22]. In resting conditions, the low frequency (LF) band reflects both sympathetic and vagal oscillations, whereas HRV oscillations in the high frequency (HF) band are exclusively linked with cardiac parasympathetic control [24–26]. Bian et al. (2016) analysed arousal in a 3D flight simulator [27], Zou et al. (2017) analysed arousal during a 3D fire evacuation, and Chittaro et al. (2017) analysed arousal in a comparison between a cemetery and a park. Finally, some studies have started to use non-linear features in emotional VR [28], as they have been shown to play a very important role in affective dynamics [10].”

 Methods:

 “Time-domain analysis includes average and standard deviation of RR intervals, the root mean square of successive differences of intervals (RMSSD) and the number of successive differences of intervals which differ by more than 50 ms (pNN50). In addition, we included the triangular interpolation of the HRV histogram and the baseline width of the RR histogram evaluated through triangular interpolation (TINN). To obtain the frequency domain features, the power spectral density (PSD) was calculated using Fast Fourier Transform and three bands: VLF (very low frequency, <0.04 Hz), LF (low frequency, 0.04-0.15 Hz) and HF (high frequency, 0.12-0.4 Hz). The peak value corresponding to the frequency having maximum magnitude and the power of the frequency band (in absolute and percentage terms) was calculated. Moreover, for the LF and HF bands, the normalised power (n.u.) was calculated as the percentage of the signals, subtracting the VLF from the total power, and the LF/HF ratio was calculated to quantify sympathovagal balance and to reflect sympathetic modulations. Finally, the total power was included. The VLF band was excluded from the analysis as it reflects changes due to slow regulatory mechanisms (e.g., thermoregulation) [24]. Moreover, Poincaré plot analysis was used; this is a quantitative-visual technique that summarises information about non-linear dynamics and detailed beat-to-beat data on heart behaviour and categorises them into functional classes: SD1, which is related with fast beat-to-beat variability, and SD2, which describes long-term variability [24].

 Many nonlinear analyses have also been performed as it has been shown they are important quantifiers of cardiovascular control dynamics mediated by the ANS in affective computing [10]. Two measures of entropy were also included: sample entropy (SampEn), which provides an evaluation of time-series regularity [38], and Approximate Entropy, which detects changes in underlying episodic behaviour not reflected in peak occurrences or amplitudes [39]. DFA correlations are divided into short-term and long-term fluctuations: α1 represents the fluctuation in the range of 4–16 samples, and α2 refers to the range of 16–64 samples [40]. Finally, the D2 feature measures the complexity of the time series, providing information on the minimum number of dynamic variables needed to model the underlying system [41].”

Q2.6) I have questions about whether any baseline physiological data were collected and how those were integrated into the analyses. There are several differences in physical movement in addition to the novelty of the VR that could account for some of the differences between physiological data patterns between the two.

R2.6) No data at baseline, i.e., without stimuli, have been collected in the frame of this research. We agree with the reviewer that this is a limitation of our study. Accordingly, we now include a new limitation statement at line 451 as follows: 

 “In addition, baseline data gathered during non-emotional elicitation should be investigated for further normalization steps aimed at improving the accuracy of the proposed computational models.”

Q2.7) Was any statistical adjustment done for multiple comparisons? There are an awful lot of analyses run in this paper.

R2.7) Thank you for this comment. As stated above, the univariate descriptive and inferential statistics was a preliminary step toward the main multifeatured-based methodological contribution, which is linked to the development of a machine learning tool for an automatic arousal recognition. 

 Following the reviewer’s suggestion, we make explicit the nature of uncorrected p-values in the new caption of Table 2, and now include a limitation statement in the Discussion section at line 393 as follows: 

 “However, the p-values of the statistical tests performed are not corrected for multiple comparisons and should be considered as a first exploratory step for the development of a multi-feature SVM for automatic arousal recognition.”

Q2.8) The authors seem to justify the investigation of VR as an emotional induction tool by suggesting that presence is required for evoking emotion, when clearly this is not the case (pg 4), in addition although the ability to simulate real-world tasks may be helpful for investigating some elements of emotion-related experience, clearly this is also not required (e.g., integral vs incidental emotion). Thus, I’m a little confused by the content on page 4. There are certainly other reasons why researchers may want to study emotion in the context of VR, so I don’t disagree that this is a worthy thing to investigate, I just find the reasoning provided to be questionable.

R2.8) We agree with the reviewer that there is no relationship between presence and emotions in the investigated scenarios. This can be seen in the case of incidental emotions that cannot be related to a specific physical stimulus. However, in the context of evoking emotions using passive stimuli, presence plays an important role since it is an indicator of the reliability of the simulation. In fact, the concept of presence, which measures the level of ‘being there’ during a simulation, have evolved in recent years, and can be divided in “place illusion” (PI) and “plausibility illusion” (PsI). While PI is related to how the world is perceived and the correlation of movements and concomitant changes in the images that form perceptions, PsI is related to what is perceived, in a correlation of external events not directly caused by the participant (Slater, 2009). PsI is determined by the extent to which a system produces events that directly relate to the participant, and the overall credibility of the scenario being depicted in comparison with viewer expectations, for example, when an experimental participant is provoked into giving a quick, natural and automatic reply to a question posed by an avatar. 

 In summary, we agree that presence is not required to evoke emotions, but we believe that it is a key measure of reliability when we are simulating environments. 

 Following the reviewer’s suggestion, the description of the additional value of using VR in affective computing research is now enriched. The revised manuscript now specifies that the importance of the presence is related to the passive stimuli, and it’s not a key factor in all the emotion elicitation method. Text at line 67 is as follows: 

 “However, these passive emotion elicitation methods have two important limitations. First, the devices usually provide 2D stimuli, which may evoke low levels of presence. Presence is the feeling of “being there” when a virtual stimulus is presented, and it is an important indicator of the simulation's reliability when we evoke emotions using passive audio-visual stimuli [14].”

Q2.9) There are several sentences that don’t make sense to me. This includes:

 Pg 3, line 46: “the applications of affective computing are impacting transversally…”

 Pg 4, line 82: “The recent technological improvements in the performance of HMDs…” – how is HMD performance boosting emotion recognition?

R2.9) Following the reviewer’s suggestions, the sentences were either removed or rephrased as follow:

 Pg 3, line 46: “with applications in healthcare [2], education [3] and entertainment [4]”

 Pg 4, line 80: “Recent technological improvements of HMDs in terms of resolution and field of view are increasing their application in many research areas, including affective computing.”

---

## [Decision Letter · Decision Letter 1]

19 May 2021

PONE-D-20-32832R1

Heart rate variability analysis for the assessment of immersive emotion elicitation using virtual reality: Comparing real and virtual scenarios in the arousal dimension

PLOS ONE

Dear Dr. Marín-Morales,

Thank you for submitting your manuscript to PLOS ONE. After careful consideration, we feel that it has merit but does not fully meet PLOS ONE’s publication criteria as it currently stands. Therefore, we invite you to submit a revised version of the manuscript that addresses the points raised during the review process.

Thank you for your revised version of your manuscript. As you can see, reviewer 2 was very pleased with your revisions and has only suggestions for some minor revisions, which I would encourage you to make.

We look forward to receiving your revised manuscript.

Kind regards,

Hedwig Eisenbarth

Academic Editor

PLOS ONE

Journal Requirements:

Reviewers' comments:

Reviewer's Responses to Questions

**Comments to the Author**

1. If the authors have adequately addressed your comments raised in a previous round of review and you feel that this manuscript is now acceptable for publication, you may indicate that here to bypass the “Comments to the Author” section, enter your conflict of interest statement in the “Confidential to Editor” section, and submit your "Accept" recommendation.

Reviewer #2: (No Response)

2. Is the manuscript technically sound, and do the data support the conclusions?

Reviewer #2: Yes

3. Has the statistical analysis been performed appropriately and rigorously? 

Reviewer #2: I Don't Know

4. Have the authors made all data underlying the findings in their manuscript fully available?

Reviewer #2: No

5. Is the manuscript presented in an intelligible fashion and written in standard English?

Reviewer #2: Yes

6. Review Comments to the Author

Reviewer #2: The authors have addressed most of my suggestions in this revision. There are two issues which remain. First, although there have been several changes to reference "arousal" rather than "emotion", i think a change in the title of the paper to this effect is also necessary.

Second, my previous comment about about validation (Q2.4 in the response to review) was not about validation of the model per se, which I understand was likely confusing due to my comment on AUC. Rather I'm hoping the authors can explain how their paper addresses the comments they raise about the "validity" of VR, such as in the abstract in line 22 and also in line 121. What does it mean that VR needs to be validated and how does the current analysis address this.

7. PLOS authors have the option to publish the peer review history of their article (what does this mean?). If published, this will include your full peer review and any attached files.

Reviewer #2: No

---

## [Author Response · Author response to Decision Letter 1]

1 Jun 2021

Reviewer #2

Comments:

Q2.1) The authors have addressed most of my suggestions in this revision. There are two issues which remain. First, although there have been several changes to reference "arousal" rather than "emotion", i think a change in the title of the paper to this effect is also necessary.

R2.1) Thank for your comment. Following the reviewer suggestion, the title has been changed to “Heart rate variability analysis for the assessment of immersive emotional arousal using virtual reality: Comparing real and virtual scenarios”

Q2.2) Second, my previous comment about validation (Q2.4 in the response to review) was not about validation of the model per se, which I understand was likely confusing due to my comment on AUC. Rather I'm hoping the authors can explain how their paper addresses the comments they raise about the "validity" of VR, such as in the abstract in line 22 and also in line 121. What does it mean that VR needs to be validated and how does the current analysis address this. 

R2.2) Thank you for your comment. Many researchers are currently using VR to replicate tasks to assess human behaviour, therefore a thorough investigation on the validity of those environments, and in particular the similitudes and differences between VR and physical environments, is critical.

 The concept of validity is presented in the introduction (line 121) and refers to the capacity of VR to evoke a response from the user equal to the one that might be evoked by a real physical environment. 

 According to such definition, the present work investigates VR validity in the emotional arousal dimension and shows that, while the direct virtualisation of a real environment might be self-reported as evoking psychological arousal, it does not necessarily evoke the same cardiovascular changes as a real arousing elicitation. 

 We believe this is an important finding, which might shed light to future VR and emotional research.

---

## [Decision Letter · Decision Letter 2]

21 Jun 2021

Heart rate variability analysis for the assessment of immersive emotional arousal using virtual reality: Comparing real and virtual scenarios

PONE-D-20-32832R2

Dear Dr. Marín-Morales,

We’re pleased to inform you that your manuscript has been judged scientifically suitable for publication and will be formally accepted for publication once it meets all outstanding technical requirements.

Kind regards,

Hedwig Eisenbarth

Academic Editor

PLOS ONE

Additional Editor Comments (optional):

The reviewer found their comments well addressed.

Reviewers' comments:

Reviewer's Responses to Questions

**Comments to the Author**

1. If the authors have adequately addressed your comments raised in a previous round of review and you feel that this manuscript is now acceptable for publication, you may indicate that here to bypass the “Comments to the Author” section, enter your conflict of interest statement in the “Confidential to Editor” section, and submit your "Accept" recommendation.

Reviewer #2: All comments have been addressed

2. Is the manuscript technically sound, and do the data support the conclusions?

Reviewer #2: Yes

3. Has the statistical analysis been performed appropriately and rigorously? 

Reviewer #2: Yes

4. Have the authors made all data underlying the findings in their manuscript fully available?

Reviewer #2: No

5. Is the manuscript presented in an intelligible fashion and written in standard English?

Reviewer #2: Yes

6. Review Comments to the Author

Reviewer #2: All review comments have been sufficiently addressed.

7. PLOS authors have the option to publish the peer review history of their article (what does this mean?). If published, this will include your full peer review and any attached files.

Reviewer #2: No

---

## [Editor Report · Acceptance letter]

23 Jun 2021

PONE-D-20-32832R2 

Heart rate variability analysis for the assessment of immersive emotional arousal using virtual reality: Comparing real and virtual scenarios 

Dear Dr. Marín-Morales:

I'm pleased to inform you that your manuscript has been deemed suitable for publication in PLOS ONE. Congratulations! Your manuscript is now with our production department. 

Kind regards, 

on behalf of

Dr. Hedwig Eisenbarth 

Academic Editor

PLOS ONE